# The Deadly Toxin Arsenal of the Tree-Dwelling Australian Funnel-Web Spiders

**DOI:** 10.3390/ijms232113077

**Published:** 2022-10-28

**Authors:** Fernanda C. Cardoso, Sandy S. Pineda, Volker Herzig, Kartik Sunagar, Naeem Yusuf Shaikh, Ai-Hua Jin, Glenn F. King, Paul F. Alewood, Richard J. Lewis, Sébastien Dutertre

**Affiliations:** 1Institute for Molecular Bioscience, The University of Queensland, St Lucia, QLD 4072, Australia; 2Centre for Excellence in Innovation of Peptide and Protein Science, The University of Queensland, St Lucia, QLD 4072, Australia; 3Brain and Mind Centre, Faculty of Medicine and Health, School of Medical Sciences, The University of Sydney, Camperdown, NSW 2006, Australia; 4Centre for Bioinnovation, University of the Sunshine Coast, Sippy Downs, QLD 4556, Australia; 5School of Science, Technology and Engineering, University of the Sunshine Coast, Sippy Downs, QLD 4556, Australia; 6Evolutionary Venomics Lab, Centre for Ecological Sciences, Indian Institute of Science, Bangalore 560012, India; 7IBMM, Université de Montpellier, CNRS, ENSCM, 34095 Montpellier, France

**Keywords:** venom, neurotoxins, ion channels, *Hadronyche*, deadliest spider, arboreal

## Abstract

Australian funnel-web spiders are amongst the most dangerous venomous animals. Their venoms induce potentially deadly symptoms, including hyper- and hypotension, tachycardia, bradycardia and pulmonary oedema. Human envenomation is more frequent with the ground-dwelling species, including the infamous Sydney funnel-web spider (*Atrax robustus*); although, only two tree-dwelling species induce more severe envenomation. To unravel the mechanisms that lead to this stark difference in clinical outcomes, we investigated the venom transcriptome and proteome of arboreal *Hadronyche cerberea* and *H. formidabilis*. Overall, *Hadronyche* venoms comprised 44 toxin superfamilies, with 12 being exclusive to tree-dwellers. Surprisingly, the major venom components were neprilysins and uncharacterized peptides, in addition to the well-known ω- and δ-hexatoxins and double-knot peptides. The insecticidal effects of *Hadronyche* venom on sheep blowflies were more potent than *Atrax* venom, and the venom of both tree- and ground-dwelling species potently modulated human voltage-gated sodium channels, particularly Na_V_1.2. Only the venom of tree-dwellers exhibited potent modulation of voltage-gated calcium channels. *H. formidabilis* appeared to be under less diversifying selection pressure compared to the newly adapted tree-dweller, *H. cerberea*. Thus, this study contributes to unravelling the fascinating molecular and pharmacological basis for the severe envenomation caused by the Australian tree-dwelling funnel-web spiders.

## 1. Introduction

Australian funnel-web spiders (FWS) are often referred to as the most dangerous spiders in the world [1,2]. The distribution of FWS is restricted to the eastern coast of Australia and coincides with areas of medium to high human population density, leading to frequent and potentially deadly encounters [2]. Severe envenomations were common before the introduction of antivenom in the 1980s, with more than a dozen human deaths recorded [3]. The reported symptoms are diverse, but the life-threatening effects mostly involve the cardio-respiratory system, including peripheral vasoconstriction, oedema, hypertension/hypotension and tachycardia [4]. Many toxins have been characterized, but the lethal components were determined to be 41–43 amino acid δ-hexatoxins, evolved for defense against predators [5].

From the most recent revision of the taxonomy of FWS, 36 species have been described from three genera based on fine anatomical differences [6,7]. The vast majority of FWS species are ground-dwelling, including the infamous *Atrax robustus* (Sydney FWS) and *Hadronyche infensa* (Toowoomba FWS); whereas, only two tree-dwelling species, *Hadronyche cerberea* (Southern tree-dwelling FWS) and *H. formidabilis* (Northern tree-dwelling FWS), have been identified. Although both are “arboreal”, there are clear behavioural differences, with *H. cerberea* building two-entrance ground-dwelling spider-like webs, mostly in decaying tree trunks (typically < 2 m high), whereas *H. formidabilis* builds webs with three to four entrances (Y or X shaped) higher up in trees. All recorded human fatalities have resulted from bites inflicted by the Sydney FWS, but other species can also lead to severe envenomation. In fact, a much higher rate of severe envenomation was reported for the tree-dwelling species compared to the ground-dwelling species, including *A. robustus* (63–75% vs. ~15–20% of bites) [1].

This bias towards more severe envenomation from arboreal species is unexplained but may arise because the tree-dwelling species tend to be larger (*H. formidabilis* is the largest species and can attain a body length of 5 cm) [6] and inject more venom. Alternatively, “dry bites” [8] may be less common in tree-dwelling species, or adaptation to a novel ecological niche (tree) may have driven the evolution of more potent venom components. Ground-dwelling FWS are known to mainly feed on insects, and to a lesser extent molluscs and annelids, whereas known predators include centipedes. While tree-dwelling species also feed primarily on insects, they have access to diverse vertebrate prey, including lizards and frogs, and may have a greater need to defend against birds and arboreal marsupials.

In this study, we investigated the venoms of the two arboreal FWS species using transcriptomics and milked venom proteomics. We also compared the effects of these venoms using blowfly toxicity and mammalian ion channel assays to better understand why tree-dwelling FWS produce more severe envenomation. Finally, we compared the influence of selection pressures on some toxin families’ rate of evolution in *H. formidabilis*, *H. cerberea* and the ground-dwelling *H. infensa*. By revealing their deadly toxin arsenal, this work provides a molecular understanding of the severe envenomation by arboreal FWS, which may improve treatment options for FWS bites.

## 2. Results

### 2.1. The Transcriptomic Diversity of H. cerberea and H. formidabilis Venom

Consistent with the recent transcriptomic analysis of *H. infensa*, the sequencing of the venom-gland transcriptomes from *H. cerberea* and *H. formidabilis* revealed a biochemically diverse venom. Expressed sequence tags were sequenced using the 454 platform and assembled using the MIRA software v.4.0 [9]. The assembly produced a total of 6685 contigs and 12,595 singlets for *H. cerberea* and 3456 contigs and 8695 singlets for *H. formidabilis*. After assembly, all contigs and singlets were submitted to the Tox|Note pipeline, and for each species, a total of 1602 and 1153 potential toxin precursors were found for *H. cerberea* and for *H. formidabilis*, respectively. Once all the sequences for both species were isolated, the previously reported Superfamily (SF) list from *H. infensa* [10] was used to get a catalogue of SF from the tree-dwelling species. We identified 33 SF for *H. cerberea* and 26 SF for *H. formidabilis*. The results show that the combined three venom transcriptomes from these *Hadronyche* FWS species express at least 44 toxin superfamilies (Figure 1, Appendix A).

Transcript abundances for each superfamily were estimated using RSEM v1.3 [11] and measured as Transcripts Per Million (TPM). Consistent with previous findings, transcript abundances amongst the three species varied greatly from low TPM counts, for example in superfamilies 11, 16 and 18, to the most abundantly expressed peptide superfamily in all three species, SF4, which, until recently, had not been previously characterised as a major component of the FWS venom arsenal [10]. Moreover, a total of 12 superfamilies were only found in *H. cerberea* or *H. formidabilis* and only four of those were shared by both species. When compared to the superfamilies in *H. infensa* venom, *H. cerberea* and *H. formidabilis* lacked 8 superfamilies (SF15, 18, 21, 22, 29, 31–33), while *H. formidabilis* lacked an additional 6 superfamilies (SF11, 19, 23–25, 27). These findings show some potential venom specialisation amongst these spider species.

The superfamilies SF9 and SF10 comprising the deadly δ-hexatoxins had a higher expression in the ground-dwelling *H. infensa* compared to the tree-dwellers, *H. cerberea* and *H. formidabilis*. A higher expression of SF1 (the double-knotting family) was found in *H. cerberea* compared to *H. infensa* and *H. formidabilis,* while a single-knotting version of SF1 was found in both tree- and ground-dwellers. In addition, tree-dwellers showed a higher expression of SF14, of which the pharmacological target is still unknown.

### 2.2. Proteomics of H. formidabilis and H. cerberea Defensive Venoms

The LC-MS analysis of the defensive venoms showed drastically different total ion count (TIC) profiles for both of the tree-dwelling species. Indeed, the venom of the southern tree-dwelling FWS appears more complex than that of its northern counterpart. In this view, the venom profile of *H. cerberea* resembles the venom profile of the ground-dwelling *H. infensa* (Appendix A), whereas *H. formidabilis* venom seems to be mostly lacking in peaks eluting >25 min (Figure 2). As already demonstrated by Palagi et al. [12], the majority of detected masses are centered around the 3–5 kDa and 7–8 kDa range.

Shotgun proteomic analysis identified 99 peptide/protein sequences in the venom of *H. formidabilis* (Appendix A) and 127 peptide/protein sequences in the venom of *H. cerberea* (Appendix A). Surprisingly, the major components (as having the most peptides matched to a sequence by PEAKS) identified in these venoms showed 68% identity with the zinc-dependent metalloprotease neprilysin previously identified in the transcriptome of the Brush-footed trapdoor spider *Trittame loki* (UniProtKB W4VS99) (Figure 3).

Other major components in these venoms included putative toxins belonging to SF2, SF4 and SF10 in the *H. cerberea* venom (Figure 4A), while the venom of *H. formidabilis* was dominated by peptides belonging to SF4 (Figure 4B). The main BLAST hits in the UniProt database for major components included U1-hexatoxin-Iw1a from *Illawarra wisharti* (Illawarra funnel-web spider) for Hc1cd and Hainantoxin-XVIII.2 from *Cyriopagopus hainanus* for Hc1j_1 in *H. cerberea*, and again Hainantoxin-XVIII.2 from *C. hainanus* for Hf1n_1 and Hf1h_1 in *H. formidabilis* (BLAST performed on 7 June 2022).

Interestingly, within SF1, we found only one peptide in *H. formidabilis* venom, while the *H. cerberea* venom contained six peptides belonging to this same family. These latest results agree with the transcriptomic findings that *H. cerberea* has a higher expression of SF1 compared to *H. formidabilis*. In addition, the proteomics agreed with the transcriptomic results by showing that *H. formidabilis* has the simplest venom as per the TIC profiling and the lower diversity in superfamilies compared to *H. cerberea* within their dominant components. These findings also support the potential venom specialization amongst these spider species.

### 2.3. Molecular Evolution of Tree-Dweller Toxins

To assess the nature and strength of the evolutionary selection that has shaped the venoms of Atracidae spiders occupying distinct ecological niches, we first employed site-specific models in CodeML of the PAML package. Our findings suggest that a few toxin superfamilies in the true tree-dwelling spiders have evolved under the strong influence of positive selection (ω: 1.5 to 2.5; positively selected sites (PS): 3 to 21), while others have experienced purifying selection (ω: 0.4 to 0.9; PS: 0 to 10). In contrast, three out of five spider toxin superfamilies from the newly adapted tree-dwelling spiders have evolved under positive selection pressure (ω: 1.8 to 2.8; PS 2 to 24), while the remaining have experienced negative selection (ω: 0.2 to 0.7; PS: 0). Similarly, toxin superfamilies, SF4, SF13 and SF26, from the ground-dwelling Atracidae spiders have experienced positive selection (ω: 1.1 to 1.5; PS: 17 to 30), while we find that SF9 and SF10 to be under influence of purifying selection (ω: 0.6 to 0.9; PS: 0 to 11; Table 1; Appendix A. Thus, our findings suggest that both positive selection and purifying selection have shaped the venom arsenal of atracid spiders. Interestingly, when MEME analyses were utilized to identify sites subjected to an episodic type of diversifying selection, we found a large number of such sites in the true tree-dwellers (0 to 45 sites) and ground-dwellers (8 to 73 sites) as compared to the newly adapted tree-dwellers (0 to 18 sites). Furthermore, when TreeSAAP analyses were used to assess the impact of molecular evolution on the biochemical and structural properties of toxins, a very large number of amino acid replacements in the newly adapted tree-dwelling and ground-dwelling spiders were identified to have a significant impact on the structure–function. In contrast, very few sites in true tree-dwelling spiders belonged to this category. Outcomes of these analyses provide evidence for a greater proportion of radical changes in newly adapted tree-dwelling and ground-dwelling spiders as compared to those of the true tree-dwelling species (Table 1 and Appendix A).

### 2.4. Insecticidal Bioassay

All FWS venoms exhibited paralytic insecticidal activity in sheep blowflies at 1–24 h post injection, except for the venom of *A. robustus* (male) at 1 h post injection (Figure 5, Appendix A). The calculated PD_50_ values (after 24 h) ranged 15-fold from 3.9 μg/g for *H. infensa* to 57.5 μg/g for *A. robustus* (male) venom. Lethal insecticidal activity was not observed for *A. robustus* female, and only 24 h post injection for the *A. robustus* male venom. Other funnel-web species were lethal and had LD_50_ values (after 24 h) that ranged 33-fold from the most potent venom at 11.5 μg/g for *H. infensa* to the weakest venom at 385 μg/g for *A. robustus* (male). These results suggest that the insecticidal effects of tree-dwelling species resemble the insecticidal effects of ground-dwelling species such as *H. infensa*. Surprisingly, the venom of the Sydney FWS *A. robustus* showed the weakest insecticidal effects and lowest PD_50_ and LD_50_ values, in relation to other FWS species.

### 2.5. Bioactivity of FWS Venoms at Mammalian Ion Channels

Venoms from the genus *Hadronyche* and one specimen of *Atrax robustus* (female) were tested on human Ca_V_ and human or rat Na_V_ channels subtypes *in vitro* (Figure 6, Appendix A). We investigated the bioactivities at 250, 25 and 2.5 ng/mL crude venom, and the results showed complex dose-dependent pharmacology with potential synergistic effects, and therefore EC_50_ values could not be calculated. Instead, radar plotting was used to better represent these data set (Figure 6). The venom of the ground-dwelling *H. infensa* strongly enhanced the activity of hNa_V_1.1, hNa_V_1.2, hNa_V_1.3, hNa_V_1.6, hNa_V_1.7, hCa_V_1.3, hCa_V_2.2 and rNa_V_1.6 at venom concentrations 250 and 25 ng/mL (Figure 6A,B). At 2.5 ng/mL, the *H. infensa* venom maintained the enhancement of activity for hNa_V_1.1, hNa_V_1.2 and hNa_V_1.3 (Figure 6C). At 250 and 25 ng/mL, the venoms from the species of tree-dwelling *H. cerberea* showed a similar trend and a consistent profile of enhancement of the responses of hNa_V_1.1, hNa_V_1.2, hNa_V_1.3, hNa_V_1.6, hNa_V_1.7, hCa_V_1.3, hCa_V_2.2 and rNa_V_1.6 (Figure 6A,B). Interestingly, at 2.5 ng/mL, these activities persisted for the sodium channels hNa_V_1.2 and hNa_V_1.3, and for the calcium channels hCa_V_1.3 and hCa_V_2.2 (Figure 6C). Similarly, the venom of the tree-dwelling *H. formidabilis* induced strong enhancement of ion channel activity at 250 and/or 25 ng/mL for the subtypes hNa_V_1.1, hNa_V_1.2, hNa_V_1.3, hNa_V_1.6, hNa_V_1.7, hCa_V_1.3, hCa_V_2.2 and rNa_V_1.6 (Figure 6A,B). At 2.5 ng/mL, the venom from *H. formidabilis* maintained activity for the sodium channels hNa_V_1.1, hNa_V_1.2, hNa_V_1.3 and hCa_V_1.3 (Figure 6C).

These results suggested that *H. cerberea* and *H. formidabilis* venoms comprise components able to enhance the activity of both sodium and calcium channels at a low concentration of 2.5 ng/mL, while the ground-dweller *H. infensa* venom was able to modulate only sodium channels at this same venom concentration. *A. robustus* (male) venom showed weak enhancement of sodium channels activity and no activity for the calcium channels tested at up to 250 ng/mL concentration (Figure 6A). These findings agreed with the results obtained from the insecticidal bioassay in which the venoms from two specimens of *A. robustus* (male and female) showed the least paralytic and lethal effects compared to the tree-dwelling *Hadronyche* species (Figure 5, Appendix A).

Interestingly, the venoms from tree-dwelling *Hadronyche* species showed a broader ion channel activity at low concentrations, which suggests that combined and potent modulation of hNa_V_1.2, hCa_V_2.2 and hCa_V_1.3 by *H. cerberea*, or hNa_V_1.2 and hCa_V_1.3 by *H. formidabilis* could be associated to more severe envenomation cases by tree-dwelling compared to ground-dwelling (Figure 7 and Appendix A).

## 3. Discussion

### 3.1. Composition and Evolution of Tree-Dwelling FWS Venoms

Like other venomous animals, investigations of the molecular diversity of spider venom components have benefited extraordinarily from combining transcriptomic and mass spectrometry methods [13]. By applying these approaches to the venom of ground- and tree-dwelling FWS, a total of 44 superfamilies of cysteine-rich venom peptides were identified in FWS. Our findings indicate a considerable diversification of the tree-dwelling spiders from their ground-dwelling counterparts, with 12 superfamilies exclusive for tree-dwelling FWS. Remarkably, the venom from the true arboreal *H. formidabilis* showed a simpler proteomic profile compared to the recently evolved tree-dwelling *H. cerberea*. In fact, the *H. cerberea* venom is more diverse and comparable to ground-dwellers; although, at a molecular level, ground-dwellers differentiate by producing higher levels of δ-hexatoxins compared to tree-dwellers. Given that bites from tree-dwelling species were more often associated with severe human envenomation, this suggests additional toxic venom peptides responsible for these effects are yet to be discovered besides the δ-hexatoxins, which likely evolved for defense against terrestrial predators [5].

The dominant presence of neprilysin-like endopeptidases in tree-dwelling FWS venoms may indicate their critical role in the envenomation mechanisms. Interestingly, earlier SDS-gel electrophoresis results showed a strong band between the two markers 68 and 116 Da, which could correspond to the size (81 kDa) of neprilysins [14]. Neprilysins have been found in other animal venoms, including other spiders, such is the case of the tarantula *Avicularia juruensis* [15], although its biological function in the venom is still unknown. These large proteins are metalloproteases that cleave at the amino side of hydrophobic residues and inactivate peptide hormones including neurotensin, oxytocin, bradykinin and angiotensin II. The identity of the *H. cerberea* neprilysin with the human ortholog is only approximately 36%, and there is no evidence the venom-derived neprilysin function as the human orthologue. Nevertheless, the prospective function of neprilysin in these venoms could likely contribute to their lethality considering the role of neprilysin in modulating key components of the hemodynamic balance, such as bradykinin and angiotensin II. The overall role of neprilysin-like endopeptidases requires further investigation to test this hypothesis.

Results of the molecular evolutionary analyses of atracid spider toxins revealed the differential influence of natural selection pressures on toxin superfamilies secreted by true tree-dwelling (*H. formidabilis*), newly adapted tree-dwelling (*H. cerberea*) and ground-dwelling (*H. infensa*) spiders. While the overall number of superfamilies under positive and negative selection pressures were very similar across the three spider groups, toxin superfamilies in the true tree-dwelling spider, *H. formidabilis*, were identified to be evolving under a relatively lesser extent of diversifying selection than their newly adapted tree- (*H. cerberea*) or ground-dwelling counterparts. In contrast, many toxin superfamilies in both the latter groups have evolved under a relatively stronger influence of positive Darwinian selection. When TreeSAAP was used to evaluate the impact of amino acid replacements on the biochemical and structural properties of toxins, a very large number of radical changes were identified in the newly adapted tree-dwelling and ground-dwelling spiders. In contrast, relatively few site replacements in the true tree-dwelling spiders were radical enough to significantly impact the structure and/or function. These findings clearly highlight the differential role of diversifying selection on the venom arsenal of the true tree-dwelling spiders, especially in comparison to the ground-dwelling spiders. When the MEME model was utilized to detect sites that may have experienced events of episodic diversifying selection, we found several of these even in the true tree-dwellers, suggesting that they too have experienced evolutionary tinkering at some point in time.

The observed differences in selection pressure experienced by toxin superfamilies of the true tree-dwelling, newly adapted tree-dwelling and ground-dwelling spiders can be attributed to the difference in the niche that these animals occupy. The true tree-dweller, *H. formidabilis*, for example, is known to occupy a higher canopy with reports of webs found as high as 18 m above ground. The newly adapted tree-dwelling *H. cerberea* species has its web in decaying tree trunks and/or the lower tree branches, much closer to the ground (<2 m), making them vulnerable to both aerial, arboreal and ground predators. Similarly, ground-dwelling atracid spiders are hunted by both aerial (e.g., parasitoid wasps, birds) and ground-dwelling predators (e.g., centipedes). This difference in predatory selection pressures could have underpinned the evolutionary expansion of the newly adapted tree-dwelling and ground-dwelling spider venom arsenal. Considering the relatively lower selection pressures exerted by a narrow range of predatory species, the venoms of the true tree-dwelling spiders may have remained largely evolutionarily constrained. Another noteworthy point concerning the variation in toxin evolution is the differences in the size of these spiders. *H. formidabilis* (the true tree-dweller) is the largest of all the Australian funnel-web spiders, ranging from 4–5 cm in body length, a size that could deter some potential predators, and could take on larger prey. In contrast, the other atracid spiders (both newly adapted tree-dweller and ground-dweller spiders) are smaller-sized and, hence, their venom toxins may experience an increased selection pressure.

### 3.2. The Insecticidal Effects of Tree-Dwelling FWS Venoms

FWS venoms are known for their potent and selective insecticidal toxins, such as ω-hexatoxins isolated from *A. robustus* venom [16], and κ- and α-hexatoxins isolated from *Hadronyche versuta* venom [17,18]. While insect Na_V_ channels are selectively targeted by spider peptides from Theraphosidae venoms [19], the FWS venoms comprise δ-hexatoxins which are highly potent to mammalian and insect Na_V_ channels, and therefore, involved in both predation of insects and defense against predators, such is the case of δ-HXTX-Ar1a isolated from *A. robustus* [5].

Our findings showed that crude venoms from tree-dwelling species resemble the insecticidal effects of ground-dwelling species such as *H. infensa*, while *A. robustus* showed a much weaker insecticidal effect. These observations urge the need to further investigate bioinsecticides from tree-dwelling FWS as well as from the ground-dwelling *H. infensa* which have not been explored at a single peptide level yet. Using an oral sheep blowfly (*Lucilia cuprina*) toxicity assay, we previously characterized crude venoms from 56 arachnid species and found that approximately 30% of these venoms were lethal [20]. In this same study the insecticidal peptides ω-Hv1a and κ-Hv1c, isolated from *H. versuta*, showed lethal effects when injected into sheep blowflies, and the oral activity of ω-Hv1a was higher than for κ-Hv1c [20]. These suggest potent bioinsecticide peptides are comprised in the venom of tree-dwellers FWS as these resemble potent insecticidal effects of ground-dwellers venoms such as from *H. infensa*.

### 3.3. Explaining the Severity of the Tree-Dwelling FWS Envenomation

FWS venoms modulate voltage-gated sodium and calcium channels, and their altered functions induce severe envenomation symptoms of the autonomic, cardiovascular, and neurological systems, as well as pulmonary oedema [1]. Ion channels in relevant physiological pathways include the subtypes hNa_V_1.5, hCa_V_1.3 and hCa_V_2.2 in the cardiac tissue; hNa_V_1.1 and hNa_V_1.6–hNa_V_1.7 in the peripheral neuronal system; hNa_V_1.1, hNa_V_1.2, hNa_V_1.6, hCa_V_1.3 and hCa_V_2.2 in the central neuronal system; hNa_V_1.1, hNa_V_1.7, hCa_V_1.3 and hCa_V_2.2 in the lung tissue; hCa_V_1.3 in the retina tissue; hNa_V_1.5 and hNa_V_1.7 in the heart and skeletal muscle tissues; hNa_V_1.2, hNa_V_1.6, hNa_V_1.7 and hCa_V_1.3 in the adrenal and pituitary glands; hCa_V_1.3, hCa_V_2.2 and hNa_V_1.7 in the skin; and hNa_V_1.6, hNa_V_1.7, hCa_V_1.3 and hCa_V_2.2 in the kidney and salivary glands [21,22]. Their localizations, previously determined by proteomic and transcriptomic studies amongst other methods [21,22], support our findings for the FWS envenomation symptoms caused by the modulation of many of these sodium and calcium channels.

Independent of their tree-or ground-dwelling lifestyle, at the highest concentration tested, all FWS venoms had a strong preference for the subtypes hNa_V_1.2 and hNa_V_1.6 in the sodium channel family and for the subtypes hCa_V_1.3 and hCa_V_2.2 in the calcium channel family, although *A. robustus* showed overall weaker activity in relation to the other FWS species. At the lowest concentration, most of these venoms were able to maintain activity at hNa_V_1.2 and hNa_V_1.3; however, tree-dwelling FWS venoms also induce potent modulation of both hCa_V_1.3 and hCa_V_2.2 while the ground-dwelling FWS venoms showed weaker modulation of hCa_V_1.3 and no activity at hCa_V_2.2. Overall, these results suggested the venom of *H. cerberea* was the most potent with activity at hNa_V_1.2, hCa_V_1.3 and hCa_V_2.2, followed by the venom of *H. formidabilis* that potently modulated hNa_V_1.2 and hCa_V_1.3, while *H. infensa* and *Atrax robustus* showed a potent modulation of hNa_V_1.2, a weak modulation of hCa_V_1.3 and did not alter hCa_V_2.2 activity. This agrees with the most recent report of clinical cases of severe envenomation in Eastern Australia at 75% for *H. cerberea*, followed by 63% for *H. formidabilis*, 17% for *A. robustus* and 14% for *H. infensa* [1].

FWS venom peptides have been classified as δ-hexatoxins, which are able to activate Na_V_ channels via the delay of channel inactivation in invertebrates and vertebrates [5,23], ω-hexatoxins, which cause invertebrate Ca_V_ channel inhibition [16,22], or λ-hexatoxins that inhibit invertebrate calcium-activated potassium channels (K_Ca_) [17]. In vertebrates, δ-HXTX-Ara1a isolated from *A. robustus* showed stronger activation of hNa_V_1.1–hNa_V_1.3 and hNa_V_1.6 compared to other hNa_V_ subtypes. Similarly, we observed a strong activity enhancement of hNa_V_1.2, hNa_V_1.3 and hNa_V_1.6 by FWS crude venoms. To date, no voltage-gated calcium channel enhancers have been described in these venoms, which indicates the potential enhancement of Ca_V_ activation responses occurs downstream of the modulation of other neuronal targets in the neuroblastoma SH-SY5Y used in these experiments such as the K_Ca_ channel [24], or a potential novel family of Ca_V_ activators in theses FWS venoms. Nonetheless, the enhancement of activity in hCsa_V_1.3 and hCa_V_2.2 channels via direct or indirect modulation remains a venom-induced mechanism that aligns with the reported severe envenomation symptoms.

Ion channel enhancers can be invaluable in the development of drugs to treat unmet neurological disorders, such as epilepsy, in which loss of function of Na_V_1.1 and Ca_V_2.2 have been demonstrated [25]. In addition to targeting voltage-gated ion channels, FWS venom comprises inhibitors of ASIC channels expressed in the cardiovascular system, such as Hi1a isolated from *H. infensa* that showed therapeutic effects in treating stroke [26,27]. As well as medical applications, new eco-friendly insecticides are described for peptides isolated from the FWS *Hadronyche versuta* [17,18] and *Atrax robusta* [16], and the commercially available bioinsecticide Spear^®^-Lep comprising ω/κ-HxTx-Hv1a from *H. versuta*.

The sodium channel subtype hNa_V_1.3 is not expressed in human adults, but it is present in the central and peripheral nervous system of small vertebrates that likely participate in the prey and feeding habits of FWS or are potential predators. We recognize the importance of the ion channel subtype hNa_V_1.4, highly expressed in skeletal muscle as a potential additional target for the severe symptoms induced by FWS venoms, although its evaluation in this work was not possible due to the limitations of the bioactivity assay with this channel subtype. In a previous study, the venom of *A. robustus* was shown to have weak activity for hNa_V_1.4 in a similar assay [5].

## 4. Materials and Methods

### 4.1. Venoms

In this study, only female tree-dwelling species and male and female *Atrax robustus* were used. Milked venom was collected directly from the fangs after the spiders were aggravated by repeatedly touching their front legs with tweezers. Pools of venom for each species were lyophilized and stored at –20 °C until use.

### 4.2. H. cerberea and H. formidabilis Library Construction and Sequencing

Two adult female *H. cerberea* spiders and one female and 3 juvenile *H. formidabilis* specimens were milked and 3 days later they were anesthetized and their dissected venom glands placed in TRIzol^®^ (Life Technologies Corporation, Carlsbad, CA, USA). Total RNA from pooled venom glands was extracted following the standard TRIzol^®^ protocol. mRNA enrichment from total RNA was performed using an Oligotex direct mRNA mini kit (Qiagen, Germantown, MD, USA). RNA quality and concentration were measured using a Bioanalyzer 2100 pico chip (Agilent Technologies, Santa Clara, CA, USA).

A cDNA library was constructed from 100 μg mRNA using the standard Roche cDNA rapid library preparation and emPCR method. Sequencing was carried out at the Australian Genome Research Facility using a ROCHE GS-FLX sequencer. The Raw Standard Flowgram File (.SFF) was converted to Fastq using the sff_extract tool in seq_crumbs (https://github.com/JoseBlanca/seq_crumbs) with default settings. Low-quality sequences were filtered using FastX-toolkit v0.0.13 (http://hannonlab.cshl.edu/fastx_toolkit/) with the following parameters: Qual cut-off value of 30 and the percent of bases in sequence value of 50. Sequences with less than 50 bp were also discarded. Reads passing QC were visualized using FastQC (https://www.bioinformatics.babraham.ac.uk/projects/fastqc/). De novo assembly was performed using MIRA software v4.0 [9] using the following parameters: -GE:not = 4 --project = Hinfensa --job = denovo, est, accurate, 454 454_SETTINGS -CL:qc = no -AS:mrpc = 1 -AL:mrs = 99, egp = 1. Each assembled dataset was analyzed by Tox|Note [28], with toxin precursors annotated in Tox|Blast and submitted to ENA (Tox|Name and Tox|Submission modules, respectively). Corresponding details for the *H. infensa* library construction, assembly and analysis are described in detail in [10]. Finally, toxin abundance in transcripts per million TPMs was estimated using RSEM v1.3.1 [11].

After analyzing and extracting all-potential full-length toxin precursors from *H. cerberea* and *H. formidabilis*, files were concatenated and a reference fasta file containing one consensus sequence from each representative superfamily (SF) was created from all species, including *H. infensa*. These two files were used to cluster all isoforms from each SF using cd-hit-2d [11] using the -c set to 0.4 and -n 2. The resulting clstr file was used to extract the ids of each SF for each species and using a regular expression, we extracted each SF isoform count from their corresponding RSEM output file. As required, manual checks were conducted to corroborate the adequate clustering of sequences. TPMs for each isoform were summarized using GraphPad Prism v7.0.0 (for MAC, GraphPad Software, San Diego, CA, USA). Toxins were named using the rational nomenclature described previously [29]. Spider taxonomy was taken from the World Spider Catalog v23.5 (https://wsc.nmbe.ch).

### 4.3. Proteomics

Liquid chromatography coupled mass spectrometry (LC-MS), shotgun proteomics and bioinformatic analysis of the results were carried out as previously described in detail [30]. Briefly, LC-MS was operated on an Acquity H-Class ultrahigh performance liquid chromatography (UPLC) system (Waters, Corp., Milford, MA, USA) fitted with a UV detector (PDA detector) under the control of Waters MassLynx software (v4.1). Separation of the crude FWS venom (300 μg) was achieved using a Phenomenex (Torrance, CA, USA) Kinetex C_18_ 100 Å column (2.1 × 150 mm, 3 µm) using a gradient of 0–80% B (0.1% formic acid in ACN) in 80 min.

For shotgun proteomics, 2.5 µL of crude FWS venom at 20.5 mg/mL in milli-Q water were diluted 20 times, reduced and alkylated with 10 µL of 500 mM iodoacetamide [30]. The venom sample was submitted to trypsin digestion and a total of 0.5 µg from the digested material was analyzed on an Acquity UPLC^®^ M-Class (Waters, Milford, MA, USA) coupled with the Q-Exactive™ Plus Hybrid Quadrupole-Orbitrap™ Mass Spectrometer (Thermo Scientific, Bremen, Germany). The chromatographic system was equipped with a monolithic PepSwift Capillary column of 100 µm × 25 cm (Thermo Scientific, Waltham, MA, USA). Peptides were eluted using a gradient of 3–50% of solution B in 100 min and 50–80% of solution B in 30 min (A: water/0.1% formic acid; B: acetonitrile/0.1% formic acid), at a flow rate of 0.6 mL/min, and data were acquired in the positive-ion mode.

PEAKS Studio 8.5 (Bioinformatics solutions, Waterloo, ON, Canada) was employed to analyze MS/MS data from *H. formidabilis* and *H. cerberea* venoms [30]. MS/MS spectra obtained from proteomic analysis were matched to our own database resulting from the assembled venom gland transcriptomes of both species of FWS translated into the six reading frames.

### 4.4. Molecular Evolution Analysis

Nucleotide datasets for toxin superfamilies 4, 9, 10, 13 and 26 were assembled from the National Center for Biotechnology Information’s non-redundant (NCBI-NR) and transcriptome shotgun assembly (TSA) databases using BLAST [31]. Translated toxin sequences were aligned in MEGA 7 using the MUSCLE algorithm [32,33] and structurally conserved cysteines were used as guides to refine the alignment.

We assessed the nature of selection shaping the evolution of toxin superfamilies using a maximum-likelihood inference implemented in CodeML of the PAML package [34]. The package estimates omega (ω) values as the ratio of non-synonymous substitutions to synonymous substitutions. We utilized the branch-site test with models M7 (Beta; null model) and M8 (Beta and ω; alternate model). A likelihood ratio test was then performed to determine the statistical significance. Posterior probabilities (P) for individual sites were calculated using the Bayes Empirical Bayes (BEB) approach implemented in M8 [35]. Sites with BEB ≥ 95% were identified to be under the influence of positive selection. The episodic nature of selection was determined using the Mixed Effect Model of Evolution (MEME) from the HyPhy package [36]. TreeSAAP (v 3.2) was employed to estimate the influence of positive selection on the biochemical and structural properties of amino acids [37]. Data acquired from TreeSAAP was further visualized and processed with IMPACT_S [37].

### 4.5. Blowfly Toxicity Assay

The toxicity of each funnel-web spider venom was determined by dissolving the venoms in an assay buffer containing (in mM): 200 NaCl, 3.1 KCl, 5.4 CaCl_2_, 4 MgCl_2_, 2 NaHCO_3_, 0.1 Na_2_HPO_4_, 0.1% bovine serum albumin (fraction V), pH 7.2, followed by injection into the ventro-lateral thoracic region of adult sheep blowflies (*Lucilia cuprina*) with average masses between 16.4 and 29.5 mg. A 1.0 mL Terumo Insulin syringe (B-D Ultra-Fine, Terumo Medical Corporation, Elkton, MD, USA) with a fixed 29 G needle fitted to an Arnold hand micro-applicator (Burkard Manufacturing Co. Ltd., Rickmansworth, UK) was used for injection with a maximum volume of 2 μL injected per fly. A total of three separate tests were carried out with three to seven doses tested per venom (n = 7 flies per dose) and the appropriate controls using PSS buffer. All flies were individually housed in 2 mL tubes and paralytic activity and lethality were determined 1 and 24 h post injection, respectively. PD_50_ and LD_50_ values were calculated, as described previously, using GraphPad Prism 6 [38].

### 4.6. Mammalian Cell Culture

Cell culture reagents were from Life Technologies Corporation, CA, USA, unless otherwise stated. The human neuroblastoma cell line SH-SY5Y was cultured in Roswell Park Memorial Institute (RPMI) medium supplemented with 15% FBS and 2 mM L-glutamine. Human embryonic kidney 293 (HEK293) cells stably expressing recombinant human (h) Na_V_ subtypes and the β1 auxiliary subunit (SB Drug Discovery, Glasgow, UK) were cultured in Minimal Essential medium (MEM) (Sigma-Aldrich, MO, USA) supplemented with 10% FBS, 100 U·mL^−1^ penicillin, 100 μg·mL^−1^ streptomycin, 2 mM L-glutamine and variable concentrations of blasticidin and geneticin according to the manufacturer’s instructions. Chinese hamster ovary (CHO) cells stably expressing recombinant hNa_V_1.6 were maintained in an F-12 medium supplemented with 10% FBS, 100 U·mL^−1^ penicillin and 100 μg·mL^−1^ streptomycin. HEK293 cells stably expressing recombinant rat (r) subtypes Na_V_1.6 and Na_V_1.7 (kindly provided by Steven Waxman at the Yale School of Medicine, New Haven, CT, USA) were cultured in Dulbecco’s Modified Eagle media (DMEM) supplemented with 10% FBS, 100 U/mL penicillin, 100 μg/mL streptomycin and geneticin. All cells were maintained at 37 °C in a humidified 5% CO_2_ incubator, and sub-cultured every 3–4 days in a ratio of 1:5 using 0.25% trypsin/EDTA.

### 4.7. Calcium Influx Assays

Venoms were screened against hNa_V_1.7 in SH-SY5Y cells using a Fluorescent Imaging Plate Reader (FLIPR Tetra; Molecular Devices, San Jose, CA, USA) as previously described [39,40]. Briefly, SH-SY5Y cells were plated at 40,000 cells per well in 384-well flat clear-bottom black plates (Corning, New York, NY, USA) and cultured at 37 °C in a humidified 5% CO_2_ incubator for 48 h before commencing assays. Cells were loaded with 20 μL per well of Calcium 4 dye (Molecular Devices) reconstituted in assay buffer containing (in mM) 140 NaCl, 11.5 glucose, 5.9 KCl, 1.4 MgCl_2_, 1.2 NaH_2_PO_4_, 5 NaHCO_3_, 1.8 CaCl_2_ and 10 HEPES pH 7.4 and incubated for 30 min at 37 °C in a humidified 5% CO_2_ incubator. For the hCa_V_1.3 assay, the N-type inhibitor CVID was added to the dye at 1 μM final concentration prior to addition to the cells. For the hCa_V_2.2 assay, the L-type inhibitor nifedipine was added to the dye at 10 μM final concentration prior to the addition to the cells. The fluorescence responses were recorded at an excitation wavelength of 470–495 nm and emission 515–575 nm for 10 s to set the baseline, 600 s after the addition of 250, 25 or 2.5 μg/mL venom, and for a further 300 s after the addition of 3 μM veratridine and 30 nM of the scorpion toxin OD1 for the hNa_V_1.7 assay, or 90 mM KCl and 5 mM CaCl_2_ for the hCa_V_ assays. Fluorescence-imaging assays used the area under the curve value after the addition of the activator KCl and CaCl_2_ for Ca_V_ channels. Due to the low number of specimens available, we did not perform statistical analysis with this set of data.

### 4.8. Membrane Potential Assays (Nav)

Changes in membrane potential were measured in HEK293 or CHO cells expressing human or rat Na_V_ subtypes with or without the β1 auxiliary subunit using an assay in which fluorescence is enhanced by membrane depolarization. Cells were plated at 10,000 cells per well in 384-well flat clear-bottom black plates (Corning) and cultured in complete media at 37 °C in a humidified 5% CO_2_ incubator for 24 h before commencing assays. For assays, cells were loaded with 20 μL per well of the red membrane potential indicator (Molecular Devices) reconstituted in the assay buffer containing (in mM) 140 NaCl, 11.5 glucose, 5.9 KCl, 1.4 MgCl_2_, 1.2 NaH_2_PO_4_, 5 NaHCO_3_, 1.8 CaCl_2_ and 10 HEPES pH 7.4, and incubated for 30 min at 37 °C in a humidified 5% CO_2_ incubator. Crude venoms were added to the wells at (in ng/mL) 250, 25 and 2.5 concentrations and plates were incubated at room temperature for 30 min before fluorescence reading. Changes in membrane potential were recorded at an excitation wavelength of 510–545 nm and emission of 565–625 nm for 10 s to set the baseline for a further 300 s after the addition of 50 μM veratridine. Fluorescence-imaging assays used the area under the curve value after the addition of the activator veratridine for Na_V_ channels. Due to the low number of specimens available, we did not perform statistical analysis with this set of data.

## 5. Conclusions

FWS venom potently modulates human voltage-gated sodium and calcium channels to induce severe envenomation symptoms and death via the fasciculation produced by the enhancement of these channel activities. Based on our results, the ion channel subtypes particularly modulated by FWS venoms and likely the major players in the deadly symptoms are Na_V_1.2, Na_V_1.6, Ca_V_1.3 and Ca_V_2.2. In tree-dweller FWS, peptides involved in such modulation belong to SF9, SF10 and SF13, and further studies testing these peptides individually will help to confirm these observations. In addition, tree-dweller venoms comprised more variants of ASIC channels peptide inhibitors. Such potent venoms probably evolved from predatory selection pressures leading the true tree-dweller *H. formidabilis* and the newly adapted tree-dweller *H. cerberea* to be named among the most dangerous venomous spider species in the world. Several spider peptides with bioinsecticide potential were unraveled in FSW venoms in this study and warrant further studies of their biotechnological potential.

## Figures and Tables

**Figure 1 ijms-23-13077-f001:**
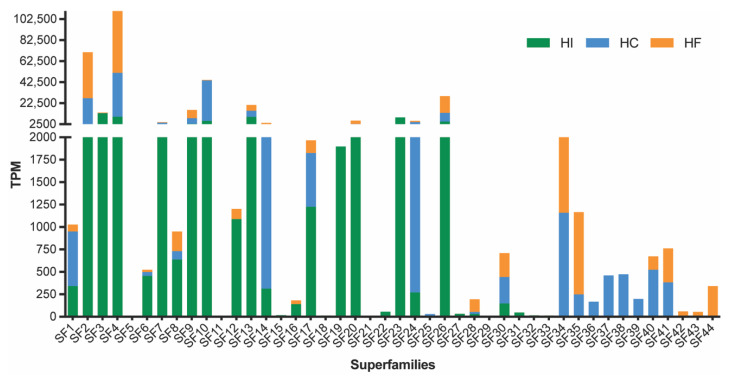
Abundance of transcripts (TPM, transcripts per million) encoding DRPs and proteins obtained from sequencing the venom-gland transcriptome of *H. cerberea* (blue bars) and *H. formidabilis* (orange bars) compared to previously published *H. infensa* transcriptome (green bars). Superfamilies were named according to the SF names previously described for *H. infensa* and included 12 novel ones.

**Figure 2 ijms-23-13077-f002:**
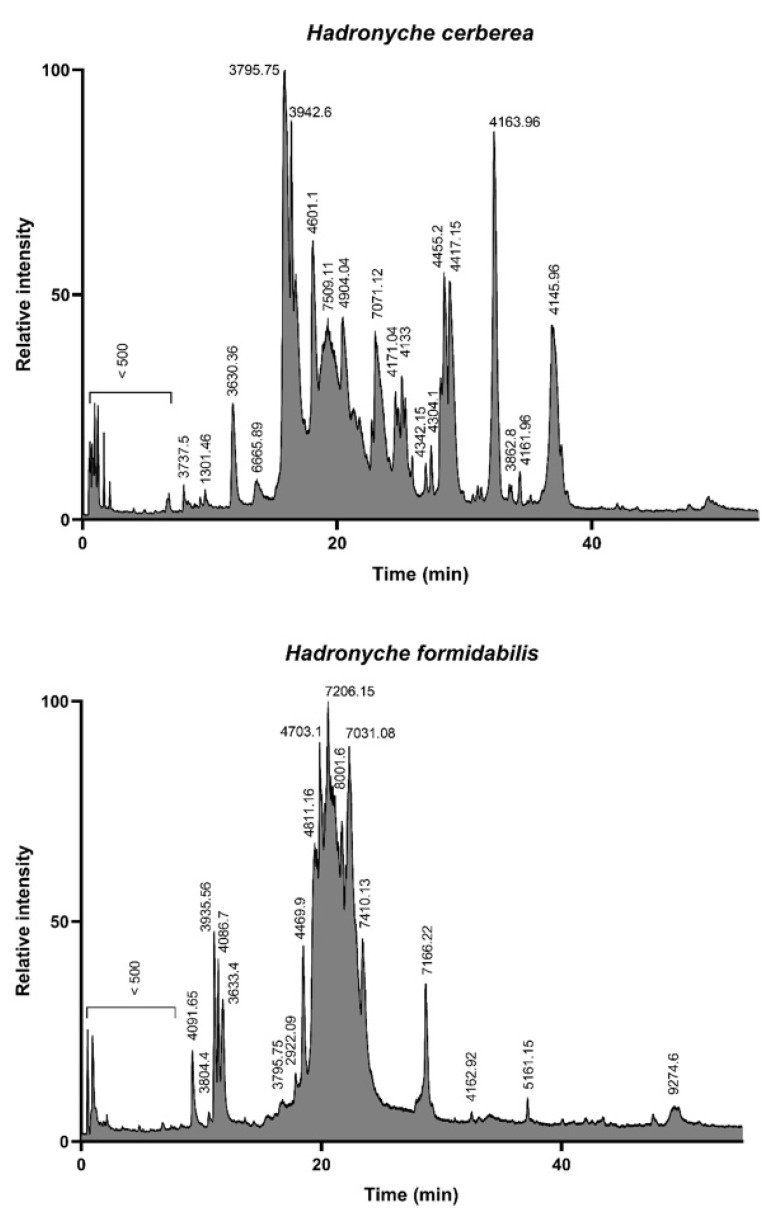
Total ion count profiles of the defensive venom of the two tree-dwelling FWS species *H. cerberea* (**top** panel) and *H. formidabilis* (**bottom** panel). The complexity of *H. cerberea* venom is reminiscent of ground-dwelling species, such as *H. infensa* (see Appendix A), with the elution of compounds spreading from 10 to 40 min, whereas *H. formidabilis* venom appears more streamlined, with most compounds eluting between 15 and 25 min.

**Figure 3 ijms-23-13077-f003:**
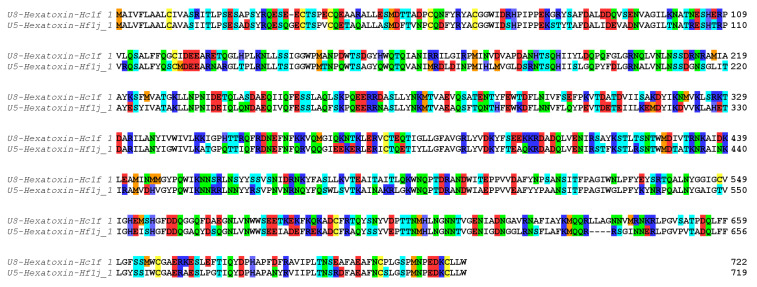
Neprilysin sequences identified in the transcriptomes and validated in the proteome of *H. cerberea* (U8-Hexatoxin-Hc1f_1) and *H. formidabilis* (U5-Hexatoxin-Hf1j_1). Color coding according to amino acid properties: blue (R, K and H), red (E and D), green (N and Q) and cyan (T and S), yellow (C), orange (M), no color (all hydrophobic residues).

**Figure 4 ijms-23-13077-f004:**
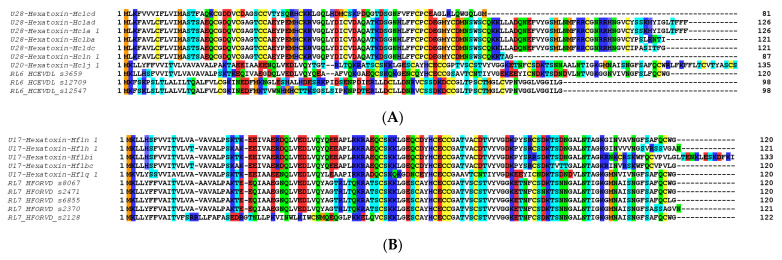
Top 10 small peptide toxin sequences identified in the venom proteome of *H. cerberea* (**A**) and *H. formidabilis* (**B**).

**Figure 5 ijms-23-13077-f005:**
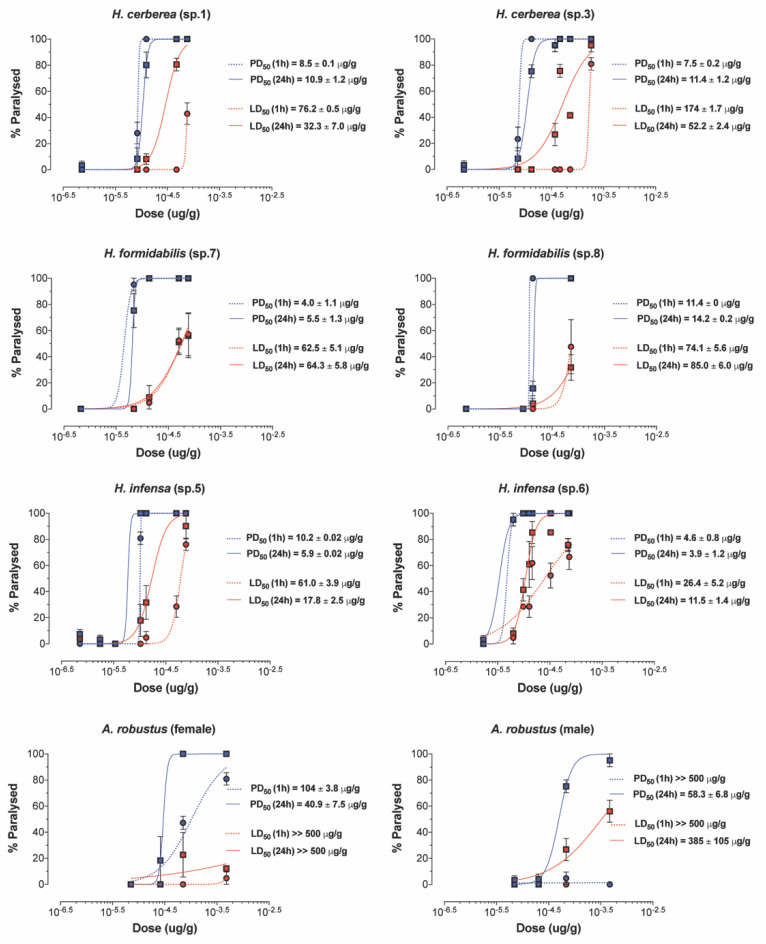
Paralytic and lethal effects of FWS venoms. Paralysis (blue) and lethality (red) were determined 1 h (dashed line) and 24 h (solid line) after injection into adult sheep blowflies (*L. cuprina*). PD_50_ and LD_50_ values are represented as mean ± SEM from n = 7 sheep blowflies tested per concentration of venom. “sp” refers to the specimen ID from which the venom was collected.

**Figure 6 ijms-23-13077-f006:**
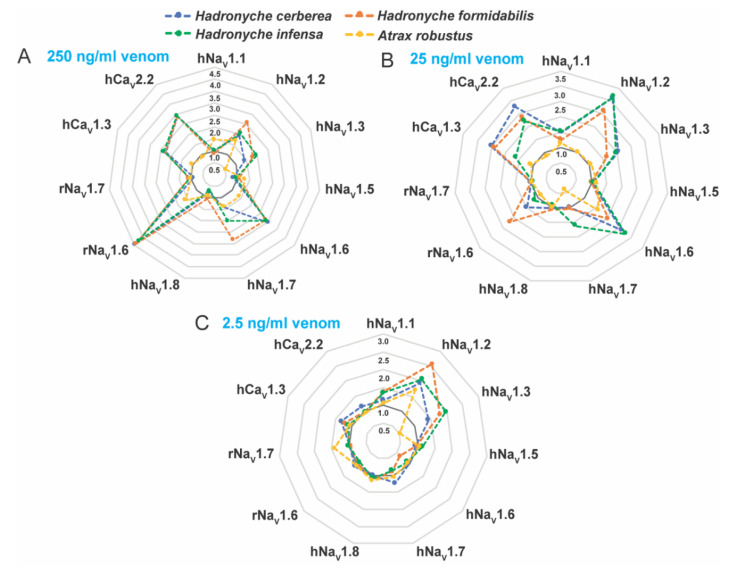
Rader plots illustrating the pharmacological activities of the FWS crude venoms at human and rat voltage-gated sodium (Na_V_) and calcium (Ca_V_) channel subtypes. Tests were performed at (**A**) 250, (**B**) 25 and (**C**) 2.5 ng/mL concentration of crude venom using fluorescence-imaging cellular assays in endogenously expressed human (h) Ca_V_ channels or recombinantly expressed rat (r) or human (h) Na_V_ channels. FWS venoms containing potent activators of these channels revealed strong preference for the subtype Na_V_1.2 in all venoms analyzed at 2.5 ng/mL. Data are represented by mean from n = 2 specimens of *H. cerbera*, *H. infensa* or *H. formidabilis*, and by n = 1 specimen of *A. robustus* for each condition tested. Ion channel activity enhancement is represented in increments of 0.5-fold from the baseline activity at 1.0 (dark grey line). Maximum fluorescence responses calculated from the area under the curve (AUC) was considered for these analyses. The standard deviation for the mean values is represented in the Appendix A.

**Figure 7 ijms-23-13077-f007:**
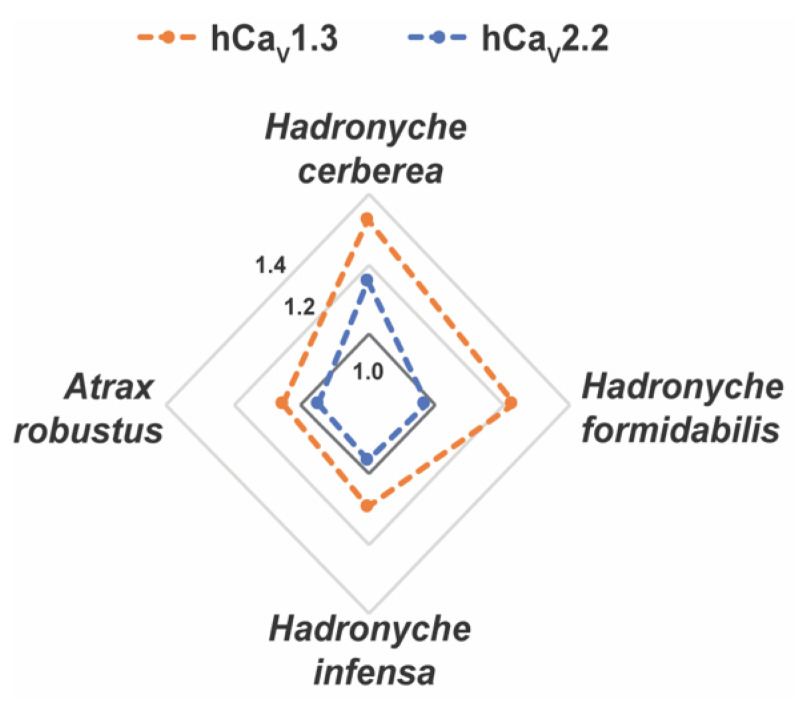
Radar plot for the pharmacological activities of the FWS crude venoms at human calcium (Ca_V_) channel subtypes. Crude venoms were tested at 2.5 ng/mL concentration using fluorescence-imaging cellular assays in endogenously expressed human (h) Ca_V_ channels in SHSY5Y neuroblastoma. Venoms from tree-dwelling FWS species *H. cerberea* and *H. fomidabilis* displayed greater enhancement of activation at hCa_V_1.3 and/or hCa_V_2.2 compared to the ground-dwelling FWS species *H. infensa* and *A. robusts* (female). Data are represented by mean from n = 2 specimens of *H. cerbera*, *H. infensa* or *H. formidales*, and by n = 1 specimen of *A. robustus* (female) for each condition tested. Channel activity enhancement are represented in increments of 0.2-fold from the baseline activity at 1.0 (dark grey line) for the Ca_V_ channels. Maximum fluorescence responses calculated from the area under the curve (AUC) were considered for these analyses.

**Table 1 ijms-23-13077-t001:** Molecular evolution of tree-dweller vs. ground-dwelling spider toxin superfamilies.

PAML (M8)			TreeSAAP						MEME Sites
Toxin Superfamily	PS	ω	Conservative Changes in A.A Properties			Radical Changes in A.A Properties			
			Chemical	Physical	Total	Chemical	Physical	Total	
*Hadronyche formidablis*: True Tree-Dweller									
SF13	-	0.64	-	-	-	-	-	-	1
SF9	-	0.62	-	-	-	-	-	-	19
SF10	3	2.59	2	-	2	-	3	3	0
SF4	10	0.98	-	16	16		51	51	45
SF26	21	1.55	-	14	14	-	53	53	6
*Hadronyche cerberea*: Newly Adapted Tree-Dweller									
SF13	-	0.20	-	-	-	-	-	-	0
SF9	2	1.90	-	-	-	-	-	-	1
SF10	-	0.75	-	-	-	-	-	-	1
SF4	23	2.85	-	19	19	-	72	72	18
SF26	24	1.85	12	6	18	-	81	81	18
*Hydronyche infensa*: Ground-Dweller from Atracidae Family									
SF13	17	1.58	-	32	32	-	7	7	8
SF9	-	0.64	-	-	-	-	-	-	15
SF10	11	0.94	-	14	14	9	-	9	8
SF4	30	1.42	-	59	79	-	327	327	73
SF26	23	1.18	-	10	10	-	119	119	14

## Data Availability

I Metadata and annotated nucleotide sequences were deposited in the European Nucleotide Archive under project accessions: (a) *H. infensa* PRJEB6062, (b) *H. cerberea* PRJEB14734, and (c) *H. formidabilis* PRJEB14965.

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
