# Peer review of "The Deadly Toxin Arsenal of the Tree-Dwelling Australian Funnel-Web Spiders"

_ijms, 2022, doi:10.3390/ijms232113077_

Round 1
Reviewer 1 Report
Dear authors,
Your manuscript provides new transcriptomic and venom proteomics information of two different arboreal spiders. The venom effect on mammalian ion channels was analysed to gain insight into the envenomation mechanism of arboreal species. The experimental work is well conducted and clear.
Discussion of results and supplementary material are sufficient. Generated data and biological activity are well integrated and discussed.
However, I would like to make a few suggestions and some remarks related to the information and analysis.
Minor remarks
Scientific names should be in italics throughout the text (A. robustus, H. infensa, H. formidabilis), lines 186, 188, 189, 191, 192, 194… 208, 212, 216… and others
The authors mention the sequence of neprilysin in figure 3. I recommend to specify which is the transcriptome-derived sequence, and which one has been found in the proteome, as a reference for mature sequences.
The toxic effect on sheep blowflies is interesting and related to the sex of the specimens. Out of curiousity, I would like to ask the authors whether they have observed paralytic or lethal effects using the venom of juvenile specimens? This observation could be added to the text.
Perhaps the authors could indicate the sex of the specimens used to collect venom in Supplementary Figure 1.
Author Response
Point-to-point responses to the reviewer’s comments
We would like to thank both reviewers for their appreciation of our work and their suggestions to improve our manuscript. Below is our reply to the queries :
Reviewer 1 : Your manuscript provides new transcriptomic and venom proteomics information of two different arboreal spiders. The venom effect on mammalian ion channels was analysed to gain insight into the envenomation mechanism of arboreal species. The experimental work is well conducted and clear. Discussion of results and supplementary material are sufficient. Generated data and biological activity are well integrated and discussed.
However, I would like to make a few suggestions and some remarks related to the information and analysis.
Minor remarks
- Scientific names should be in italics throughout the text (A. robustus, H. infensa, H. formidabilis), lines 186, 188, 189, 191, 192, 194… 208, 212, 216… and others
Thank you, we have now italicized all scientific names in our manuscript.
- The authors mention the sequence of neprilysin in figure 3. I recommend to specify which is the transcriptome-derived sequence, and which one has been found in the proteome, as a reference for mature sequences.
We realized this was confusing from the title of the legend. Actually, both transcriptomic sequences shown in figure 3 were found in the proteomes. To make it clearer, we have now explicitly indicated « Neprilysin sequences identified in the transcriptomes and validated in the proteome of H. cerberea (U8-Hexatoxin-Hc1f_1) and H. formidabilis (U5-Hexatoxin-Hf1j_1) »
- The toxic effect on sheep blowflies is interesting and related to the sex of the specimens. Out of curiousity, I would like to ask the authors whether they have observed paralytic or lethal effects using the venom of juvenile specimens? This observation could be added to the text.
Ontogenic venom differences are certainly an interesting topic on its own, but specifically to avoid such venom variability issues (as well as between male and female) we have only collected the venom from adult females, and not juvenile specimens.
- Perhaps the authors could indicate the sex of the specimens used to collect venom in Supplementary Figure 1.
Thank you, we have now mentioned the sex of the specimens (all females) from which the venom was collected in Suppl Fig 1.
Reviewer 2 Report
Article Title: The deadly toxin arsenal of the tree-dwelling Australian funnel-web spiders
In this article, the authors describe - by a combination of transcriptomic, proteomic and several in vitro and in vivo activity assays – the profile of toxins present in the venom of two tree-dewlling Australian spiders. The work is an extensive, well-detailed and rigorous characterization of venoms from H. cerberea and H. formidabilis in which over a hundred of bioactive molecules exist. The work substantially helps to unravel the toxic mechanisms associated with accidents caused by these spider’s bite. Moreover, it helps to uncover how evolutionary forces shaped the capability of these animals to stablish on their ecological niche. The overall opinion of this reviewer regarding the recommendation of this paper for publication could not be different from a resounding YES!
The notes presented bellow correspond to minor points that came out while reading this very interesting paper. If authors think they are valuable, it should be incorporated to the work in order to make it even more attractive for the readers.
- Although there is a straight relationship between the severity of symptoms after envenomation and the activity of venom components at ion channels, it is tricky to argue that the severity is based mainly on channel activation caused by the venom. In fact, ion channels inhibition importantly accounts for the severity of the symptoms as well and the final outcome is a result of net activation/inhibition of different targets. Moreover, neurotoxic symptoms are not only restricted to Nav and Cav voltage-gated channels but also on several other classes including ligand-gated channels, temperature-gated, potassium channels, ASICS, and so on. The Discussion section of the work would benefit from a more narrative presentation of the data rather than trying to stablishing a cause-consequence relationship on this.
- TPM (transcript per million) should be detailed right on the figure legend as it appears for the 1st time on the text.
- Lines 132-134. Authors should describe what “major components” really means. Alternatively, they should explain how these major components were depicted from the whole venom to perform the similarity analysis.
- Figure 5. Please describe what Sp.1 sp.3, sp.5, etc means.
- Methods section. Topic 4.8 describes membrane potential assays. Does this methodology refers to Nav assays? If so, to the name of this topic could be re-writen.
- Conclusion. The following contents: “… tree-dweller venoms comprised more variants of ASIC channels peptide inhibitors. (Lines 557-558)” and the second paragraph of conclusion resembles more a discussion of the results rather than a conclusion from this work based on the presented data. Authors should consider drawing a conclusion more concise and focused on the work’s aim.
Author Response
Point-to-point responses to the reviewer’s comments
We would like to thank both reviewers for their appreciation of our work and their suggestions to improve our manuscript. Below is our reply to the queries :
Reviewer 2 : In this article, the authors describe - by a combination of transcriptomic, proteomic and several in vitro and in vivo activity assays – the profile of toxins present in the venom of two tree-dewlling Australian spiders. The work is an extensive, well-detailed and rigorous characterization of venoms from H. cerberea and H. formidabilis in which over a hundred of bioactive molecules exist. The work substantially helps to unravel the toxic mechanisms associated with accidents caused by these spider’s bite. Moreover, it helps to uncover how evolutionary forces shaped the capability of these animals to stablish on their ecological niche. The overall opinion of this reviewer regarding the recommendation of this paper for publication could not be different from a resounding YES!
The notes presented bellow correspond to minor points that came out while reading this very interesting paper. If authors think they are valuable, it should be incorporated to the work in order to make it even more attractive for the readers.
- Although there is a straight relationship between the severity of symptoms after envenomation and the activity of venom components at ion channels, it is tricky to argue that the severity is based mainly on channel activation caused by the venom. In fact, ion channels inhibition importantly accounts for the severity of the symptoms as well and the final outcome is a result of net activation/inhibition of different targets. Moreover, neurotoxic symptoms are not only restricted to Nav and Cav voltage-gated channels but also on several other classes including ligand-gated channels, temperature-gated, potassium channels, ASICS, and so on. The Discussion section of the work would benefit from a more narrative presentation of the data rather than trying to stablishing a cause-consequence relationship on this.
We agree with the reviewer that venoms in general, including some spider venoms, contain a complex mixture of neurotoxins for which the mode of action is often not known. Yet, many act as ion channel inhibitors indeed (i.e. the paralytic alpha-neurotoxins from snake venoms or cone snail venoms), but other modulators have been reported, including activators. In the case of FWS, the contribution of activators (e.g. such as delta-HXTXs) towards human envenomation symptoms is well established in the literature. Some of the inhibitors such as the kappa and omega-HXTXs are selective towards invertebrate targets and not active against vertebrates. We not aware of any studies reporting the effects on FWS inhibitor toxins on humans. In addition, the ASIC toxins are only expressed at rather low quantities, making it unlikely that they could contribute anything significant towards the symptoms observed in humans. So overall, given that there is no real good evidence that the inhibitors actually contribute towards the symptoms in humans, we believe that it would be too speculative for us to mention in the discussion.
- TPM (transcript per million) should be detailed right on the figure legend as it appears for the 1st time on the text.
Thank you, we agree and have now added the definition of TPM in the legend of figure 1.
- Lines 132-134. Authors should describe what “major components” really means. Alternatively, they should explain how these major components were depicted from the whole venom to perform the similarity analysis.
Thank you for this remark. We define as « major components » the sequences for which PEAKS matched the largest number of peptides from MS/MS experiments. Although MS is not a quantitative method, the number of peptide matched to a particular sequence can generally be used as a proxy for relative abundance (see for instance the method of Zybailov et al., J Protome Res, 2011). This is now stated page 4 line 136.
- Figure 5. Please describe what Sp.1 sp.3, sp.5, etc means.
We have now specified in the legend of figure 5 « “sp” refers to the specimen ID from which the venom was collected ».
- Methods section. Topic 4.8 describes membrane potential assays. Does this methodology refers to Nav assays? If so, to the name of this topic could be re-writen.
Yes, correct. We have now specified this in the title of the method section 4.8.
- Conclusion. The following contents: “… tree-dweller venoms comprised more variants of ASIC channels peptide inhibitors. (Lines 557-558)” and the second paragraph of conclusion resembles more a discussion of the results rather than a conclusion from this work based on the presented data. Authors should consider drawing a conclusion more concise and focused on the work’s aim.
We agree and have re-arranged the conclusion to be more focused on our study’s aim.
